Genome-wide identification and characterization of WRKY gene family in Salix suchowensis

http://orcid.org/0000-0002-6951-2464 Bi Changwei 1
http://orcid.org/0000-0002-7962-6668 Xu Yiqing 1
Ye Qiaolin 1
Yin Tongming 2
Ye Ning 1 yening@njfu.edu.cn
1 College of Information Science and Technology, Nanjing Forestry University , Nanjing, Jiangsu , China
2 College of Forest Resources and Environment, Nanjing Forestry University , Nanjing, Jiangsu , China
Choi Doil
Electronic publication date: 2016 Sep 7
Publication date: 2016
Volume: 4
Electronic Location ID: e2437
Received 2016 Jun 24; Accepted 2016 Aug 13
Copyright: © 2016 Bi et al.
Copyright year: 2016
Copyright holder: Bi et al.
License: This is an open access article distributed under the terms of the Creative Commons Attribution License, which permits unrestricted use, distribution, reproduction and adaptation in any medium and for any purpose provided that it is properly attributed. For attribution, the original author(s), title, publication source (PeerJ) and either DOI or URL of the article must be cited.
License URL: https://creativecommons.org/licenses/by/4.0/

Keywords: Expression, Evolution, Phylogenetic analysis, Willow, Duplication, WRKY protein

Funding: Fundamental Research Funds for the Central Non-profit Research Institution of CAF CAFYBB2014QB015 National Basic Research Program of China (973 Program) 2012CB114505 National Natural Science Foundation of China 31570662, 31500533 and 61401214 National Science & Technology Pillar Program during the Twelfth Five-year Plan Period 2012BAD01B07 Natural Science Foundation of the Jiangsu Higher Education Institutions 14KJB520018 This study was supported by the Fundamental Research Funds for the Central Non-profit Research Institution of CAF (CAFYBB2014QB015), National Basic Research Program of China (973 Program) (2012CB114505) and the National Natural Science Foundation of China (31570662, 31500533 and 61401214). We also received support from Key Projects in the National Science & Technology Pillar Program during the Twelfth Five-year Plan Period (No. 2012BAD01B07), and Natural Science Foundation of the Jiangsu Higher Education Institutions (14KJB520018). This work is also enabled by the Innovative Research Team Program of the Educational Department of China, the Innovative Research Team Program in Universities of Jiangsu Province, Scientific Research Foundation for Advanced Talents and Returned Overseas Scholars of Nanjing Forestry University and the PAPD (Priority Academic Program Development) program at Nanjing Forestry University. The funders had no role in study design, data collection and analysis, decision to publish, or preparation of the manuscript.

==============================
WRKY proteins are the zinc finger transcription factors that were first identified in plants. They can specifically interact with the W-box, which can be found in the promoter region of a large number of plant target genes, to regulate the expressions of downstream target genes. They also participate in diverse physiological and growing processes in plants. Prior to this study, a plenty of WRKY genes have been identified and characterized in herbaceous species, but there is no large-scale study of WRKY genes in willow. With the whole genome sequencing of Salix suchowensis, we have the opportunity to conduct the genome-wide research for willow WRKY gene family. In this study, we identified 85 WRKY genes in the willow genome and renamed them from SsWRKY1 to SsWRKY85 on the basis of their specific distributions on chromosomes. Due to their diverse structural features, the 85 willow WRKY genes could be further classified into three main groups (group I–III), with five subgroups (IIa–IIe) in group II. With the multiple sequence alignment and the manual search, we found three variations of the WRKYGQK heptapeptide: WRKYGRK, WKKYGQK and WRKYGKK, and four variations of the normal zinc finger motif, which might execute some new biological functions. In addition, the SsWRKY genes from the same subgroup share the similar exon–intron structures and conserved motif domains. Further studies of SsWRKY genes revealed that segmental duplication events (SDs) played a more prominent role in the expansion of SsWRKY genes. Distinct expression profiles of SsWRKY genes with RNA sequencing data revealed that diverse expression patterns among five tissues, including tender roots, young leaves, vegetative buds, non-lignified stems and barks. With the analyses of WRKY gene family in willow, it is not only beneficial to complete the functional and annotation information of WRKY genes family in woody plants, but also provide important references to investigate the expansion and evolution of this gene family in flowering plants.

Introduction

Plants form a series of adjustment mechanisms to adapt diverse environment stress in their long evolutionary processes. Among the numerous adjustment mechanisms, transcription factors play important roles (Jang, Choi & Hwang, 2010). In plants, WRKY proteins constitute a large family of transcription factors, involving in various physiological and developmental processes (Eulgem, 2000; Rushton et al., 2010). Since the first WRKY gene was cloned and characterized from sweet potato (Ishiguro & Nakamura, 1994), many corresponding studies have been conducted rapidly, such as Arabidopsis thaliana, desert legume (Retama raetam), cotton (Gossypium arboreum), rice (Oryza sativa), Pinus monticola, barley (Hordeum vulgare), sunflower, cucumber (Cucumis sativus), poplar (Populus trichocarpa), tomato (Solanum lycopersicum) and grapevine (Vitis vinifera) (Ding et al., 2015; Eulgem, 2000; Giacomelli et al., 2010; Guo et al., 2014; He et al., 2012; Huang et al., 2012; Ling et al., 2011; Liu & Ekramoddoullah, 2009; Mangelsen et al., 2008; Pnueli et al., 2002; Wu, 2005).

The existence of either one or two highly conserved WRKY domains is the most vital structural characteristic of WRKY gene. WRKY gene consists of about 60 amino acid residues with a conserved WRKYGQK heptapeptide at its N-termini, and a zinc finger motif (C-X4–5-C-X22–23-H-X1-H or C-X7-C-X23-H-X1-C) at the C-terminal region. Previous functional studies indicated that WRKY genes could specifically interact with the W-box ([C/T]TGAC[T/C]), the promoter region of plant target genes, to adjust the expressions of downstream target genes (Ciolkowski et al., 2008). Additionally, SURE (sugar responsive elements), another prominent cis-element that can promote transcription processes, was also found to bind to the WRKY transcription factors under a convincing research (Sun, 2003). The proper DNA-binging ability of WRKY genes could be influenced by the variation of the conserved WRKYGQK heptapeptide (Duan et al., 2007; Maeo et al., 2001).

The WRKY proteins can be classified into three main groups (I, II and III) on the basis of the number of their WRKY domains and the pattern of the zinc finger motif. Proteins from group I contain two WRKY domains followed by a C2H2 zinc finger motif, while the other WRKY proteins from group II and III only contain one WRKY domain followed by a C2H2 or C2HC correspondingly (Yamasaki et al., 2005). Group II can be further divided into five subgroups from IIa to IIe based on additional amino acid motifs present outside the WRKY domain. Apart from the conserved WRKY domains and the zinc finger motif, there are some WRKY proteins appearing to have basic nuclear localization signal, leucine zipper (LZs) (Cormack et al., 2002), serine-threonine-rich region, glutamine-rich region and proline-rich region (Ülker & Somssich, 2004). Throughout the studies of WRKY gene family in many higher plants (Liu & Ekramoddoullah, 2009; Rushton et al., 2010; Wu, 2005), WRKY genes have been identified to be involved in various regulatory processes mediated by different biotic and abiotic stresses (Ramamoorthy et al., 2008). In plant defense against various biotic stresses, such as bacterial, fungal and viral pathogens, it has been well documented that the WRKY genes play vital roles (Cheng et al., 2015; Dong, Chen & Chen, 2003; He et al., 2016; Jaffar et al., 2016; Jiang et al., 2016; Kim et al., 2016; Li et al., 2006; Liu et al., 2016; Xu et al., 2006; Zhou et al., 2008). They are also involved in abiotic stress-induced gene expression. In Arabidopsis, with the either heat or salt treatments, the expressions of AtWRKY25 and AtWRKY33 are transformed apparently (Jiang & Deyholos, 2009). Furthermore, the expression of TcWRKY53 that belonged to alpine penny grass (Thlaspi caerulescens) is affected by salt, cold, and polyethylene glycol treatments (Wei et al., 2008). In rice, a total of 54 OsWRKY genes showed noticeable differences in their transcript abundance under the abiotic stress such as cold, drought, and salinity (Ramamoorthy et al., 2008). There is also accumulating evidence that WRKY genes are involved in regulating developmental processes, such as embryo morphogenesis (Lagacé & Matton, 2004), senescence (Robatzek & Somssich, 2002), trichome initiation (Johnson, Kolevski & Smyth, 2002), and some signal transduction processes mediated by plant hormones including gibberellic acid (Zhang et al., 2004), abscisic acid (Zou et al., 2004), or salicylic acid (Du & Chen, 2008).

The number of WRKY genes in different species varies tremendously. For instance, there are 72 members in Arabidopsis thaliana, at least 45 in barley, 57 in cucumber, 58 in physic nut (Jatropha curcas), 59 in grapevine, 104 in poplar, 105 in foxtail millet (Setaria italica), 112 in Gossypium raimondii and more than 109 in rice (Ding et al., 2015; Eulgem, 2000; Guo et al., 2014; He et al., 2012; Ling et al., 2011; Mangelsen et al., 2008; Muthamilarasan et al., 2015; Wu, 2005; Xiong et al., 2013). Zhang & Wang (2005) also identified the most basal WRKY genes in the lineage of non-plant eukaryotes and green alga. Interestingly, the WRKY genes in eukaryotic unicellular chlamydomonas, protoctist (Giardia lambliad), bryophyte (Physcomitrella patens) and fern (Ceratopteris richardii) all belonged to group I (Yu, Chen & Zhang, 2006; Ülker & Somssich, 2004; Zhang & Wang, 2005). For example, the study in bryophyte (Physcomitrella patens) found at least 12 WRKY genes, and all the genes belonged to group I (Ülker & Somssich, 2004). Additionally, the study in gymnosperm (Cycas revolute) identified at least 21 WRKY genes (Yu, Chen & Zhang, 2006), and they were divided into two groups, 15 WRKY genes therein belonged to group I and the other six WRKY genes belonged to group II. Further study suggested that the core WRKY domains of group II and III were similar to the C-terminal domain of group I; therefore, the group II WRKY genes might emerge from the breakage of the C-terminal domain in group I and the group III probably evolve from group II (Ülker & Somssich, 2004). All the above studies indicated that the group I WRKY genes might be the oldest type, which evolved from the origin of eucaryon, and group II and III might generate after the origin of bryophyte (Xie et al., 2005; Zhang & Wang, 2005). In the evolution of WRKY genes, gene duplication events played prominent roles. As a matter of fact, gene duplication events can lead to the generation of new genes. For example, there are approximately 80% of OsWRKY (rice) genes located in duplicated regions (Wu, 2005), as well as 83% of PtWRKY (poplar) genes (He et al., 2012). However, no gene duplication events have occurred in cucumber (Ling et al., 2011).

In the last few years, the increasing consumption of fossil fuels induced in a substantial increase of CO2 concentration, which has adverse impacts on global climate changes (Pleguezuelo et al., 2015). Therefore, an ever-increasing demand for energy from renewable sources has provided a new impetus to cultivate woody plants for bioenergy production. Due to its ease of propagation, rapid growth and high yield on short rotation systems, some willow species have been used as renewable resources since the 1970s. Additionally, with its essential physiological characteristics, willow becomes a prominent part of basket production, environmental restoration, analgesic extraction, phytoremediation, both riparian and upland erosion control and biomass production (Kuzovkina & Quigley, 2005). WRKY proteins participate in diverse physiological and developmental processes in plants. With these various important factors and the recent released Salix suchowensis genome sequence, which covers about 96% of the expressed gene loci (Dai et al., 2014), we have the opportunity to analyze the willow WRKY gene family. The characterization of WRKY genes in willow can provide interesting gene pools to be investigated for breeding and genetic engineering purposes in woody plants.

Materials and Methods

Datasets and sequence retrieval

The sequence of a shrub willow Salix suchowensis (S. suchowensis), which flowers within two years, was conducted with a combined approach using Roche/454 and Illumina/HiSeq-2000 sequencing technologies (Dai et al., 2014). The latest v5.2 S. suchowensis genome annotation information (version5_2.gff3) and protein sequences (Willow.gene.pep) were downloaded from our laboratory website (http://bio.njfu.edu.cn/ss_wrky/). Sequences of 72 Arabidopsis WRKY proteins were obtained from TAIR (release 10; http://www.arabidopsis.org/), and 104 poplar WRKY proteins were obtained from the Supplemental Information 3 file of poplar in He et al. (2012) (Eulgem, 2000).

Identification and distribution of WRKY genes in willow

The procedure performed to identify putative WRKY proteins in willow was similar to the method described in other species (Guo et al., 2014; He et al., 2012; Wu, 2005). The Hidden Markov Model (HMM) profile for the WRKY transcription factor was downloaded from the Pfam database (http://pfam.xfam.org/) with the keyword ‘PF03106’ (Punta et al., 2012). The HMM profile was applied as a query to search against the all willow protein sequences (Willow.gene.pep) using BLASTP program (E-value cutoff = 1e-3) (Camacho et al., 2009). Another procedure was performed to validate the putative accuracy. An alignment of WRKY seed sequences in Stockholm format from Pfam database was used by HMMER program (hmmbuild) to build a HMM model, and then the model was used to search the willow protein sequences by another HMMER program (hmmsearch) with default parameters (Eddy, 1998). Finally, we employed the SMART program (http://smart.embl-heidelberg.de/) to confirm the candidates from the two procedures correlated with the WRKY structure features (Letunic, Doerks & Bork, 2015).

Additionally, we calculated the length, molecular weight (MW), isoelectric point (PI) of these putative WRKY proteins by ExPasy site (http://au.expasy.org/tools/pi_tool.html). Every WRKY genes were mapped onto chromosomes (http://bio.njfu.edu.cn/ss_wrky/version5_2.fa) with an in-house Perl script (http://bio.njfu.edu.cn/willow_chromosome/BuildGff3_Chr.pl), and then renamed based on their orderly given chromosomal distribution. The distribution graph of every WRKY gene was drawn by MapInspect software (http://mapinspect.software.informer.com/).

Sequence alignments, phylogenetic analysis and classification of willow WRKY genes

Using the online tool SMART, we obtained the conserved WRKY core domains of predicted SsWRKY genes, and then multiple sequence alignment based on these domains was performed using ClustalX (version 2.1) (Larkin et al., 2007). After alignment, we used Boxshade (http://www.ch.embnet.org/software/BOX_form.html) to color the alignment result online. To gain a better classification of these SsWRKY genes, a further multiple sequence alignment including 103 SsWRKY domains and 82 WRKY domains from Arabidopsis (AtWRKY) was performed using ClustalW (Larkin et al., 2007), and a phylogenetic tree based on this alignment was built by MEGA 6.0 with the Neighbor-joining (NJ) method (Tamura et al., 2013). Bootstrap values have been calculated from 1,000 iterations in the pairwise gap deletion mode, which is conducive to the topology of the NJ tree by divergent sequences. Based on the phylogenetic tree constructed by SsWRKY and AtWRKY domains, these SsWRKY genes were classified into different groups and subgroups. In order to get a better comparison of WRKY family in Salicaceae, a phylogenetic tree including all SsWRKY domains and 126 WRKY domains from poplar (PtWRKY) was constructed with the similar method to Arabidopsis. Additionally, a phylogenetic tree based on full-length SsWRKY genes was also constructed to get a better classification. The ortholog of each SsWRKY gene in Arabidopsis and poplar was based on the phylogenetic trees of their respective WRKY domains, and the members of group I WRKY genes were considered as orthologs unless the same phylogenetic relationship can be detected between N-termini and C-termini in the tree. Another method described by Zou et al. (2016), BLAST-based method (Bi-direction best hit), was used to verify the putative orthologous genes (E-value cutoff = 1e-20) (Chen et al., 2007).

Evolutionary analysis of WRKY III genes in willow

The group of WRKY III genes, only found in flowering plants, is considered as the evolutionary youngest groups, and plays crucial roles in the process of plant growth (He et al., 2012; Wu, 2005). As described by Wang et al. (2015), the WRKY III genes also have a prominent impact on disease and drought resistance. Previous study of Zhang & Wang (2005) held the opinion that duplications and diversifications were plentiful in WRKY III genes, and they appeared to have confronted different selection challenges. Phylogenetic analysis of WRKY III genes was performed using MEGA6.0 with 65 WRKY III genes from Arabidopsis (AtWRKY), Populus (PtWRKY), grape (VvWRKY), willow (SsWRKY) and rice (OsWRKY). A NJ tree was constructed with the same method described before. Additionally, we estimated the non-synonymous (Ka) and synonymous (Ks) substitution ratio of SsWRKY III genes to verify whether selection pressure participated in the expansion of SsWRKY III genes. Each pair of these WRKY III protein sequences was first aligned using ClustalW. The alignments generated by ClustalW and the corresponding cDNA sequences were submitted to the online program PAL2NAL (http://www.bork.embl.de/pal2nal/) (Suyama, Torrents & Bork, 2006), which automatically calculates Ks and Ka by the codeml program in PAML (Yang, 2007).

Analysis of exon–intron structure, gene clusters, gene duplication events and conserved motif distribution of willow WRKY genes

The exon–intron structures of the willow WRKY genes were obtained based on the protein annotation files assembled ourselves (http://bio.njfu.edu.cn/ss_wrky/version5_2.gff3), and the diagrams were obtained from the online website Gene Structure Display Server (GSDS: http://gsds.cbi.pku.edu.cn/) (Hu et al., 2015).

Gene clusters are very important for predicting co-expression genes or potential function of clustered genes in angiosperms (Overbeek et al., 1999). They can be defined as a single chromosome containing two or more genes within 200 kb (He et al., 2012; Holub, 2001).

Gene duplication events were always considered as the vital sources of biological evolution. Two or more adjacent homologous genes located on a single chromosome were considered as tandem duplication events (TDs), while homologous gene pairs between different chromosomes were defined as SDs (Liu & Ekramoddoullah, 2009). BLASTP (E-value cutoff = 1e-20) was performed to identify the gene duplication events in SsWRKY genes with the following definition (Gu et al., 2002; He et al., 2012): (1) the coverage of the aligned sequence ≥ 80% of the longer gene; and (2) the similarity of the aligned regions ≥ 70%. In this study, we set the cutoff of the similarity of the aligned regions as 65%, because the similarity of the unaligned regions may reduce the value in different species.

To better exhibit the structural features of SsWRKY proteins, the online tool MEME (Multiple Expectation Maximization for Motif Elicitation) was used to identify the conserved motifs in the encoded SsWRKY proteins (Bailey et al., 2006). The optimized parameters were employed as the following: any number of repetitions, maximum number of motifs = 20, and the optimum width of each motif was constrained to between 6–50 residues. The online program 2ZIP (http://2zip.molgen.mpg.de/) was used to verify the existence of the conserved Leu zipper motif (Bornberg-Bauer, Rivals & Vingron, 1998), whereas some other important conserved motifs, HARF, LXXLL (X, any amino acid) and LXLXLX, were identified manually.

Expression analyses of willow WRKY genes

The sequenced S. suchowensis RNA-HiSeq reads from five tissues including tender roots, young leaves, vegetative buds, non-lignified stems and barks generated in our previous study were separately mapped back onto the SsWRKY gene sequences using BWA (mismatch ≤ 2 bp, other parameters as default) (Li & Durbin, 2009), and the number of mapped reads for each WRKY gene was counted. Normalization of the mapped reads was done using RPKM (reads per kilo base per million reads) method (Wagner, Kin & Lynch, 2012). The heat map for tissue-specific expression profiling was generated based on the log2 RPKM values for each gene in all the tissue samples using R package (Gentleman et al., 2004).

Results

Identification and characterization of 85 WRKY genes in willow (Salix suchowensis)

In this study, we obtained 92 putative WRKY genes by using HMMER to search the HMM profile of WRKY DNA-binding domain against willow protein sequences, and validated the accuracy of the consequence by BLASTP. After submitting the 92 putative WRKY genes to the online program SMART, seven genes without a complete WRKY domain were removed, while the other 85 WRKY genes were selected as possible members of the WRKY superfamily.

WRKY genes contain one or two WRKY domains, comprising a conserved WRKYGQK heptapeptide at the N-termini and a novel zinc finger motif (C-X4–7-C-X22–23H-X-H/C) at the C-termini (Eulgem, 2000). The variations of WRKY core domain or zinc finger motif may lead to the binding specificities of WRKY genes, but this remains to be largely demonstrated (Brand et al., 2013; Rinerson et al., 2015; Yamasaki et al., 2005). In order to identify the variations in WRKY core domains, a multiple sequence alignment of 85 SsWRKY core domains was conducted, and the result was shown in Fig. 1. Among the selected 85 WRKY genes, 81 (95.3%) were identified to have highly conserved sequence WRKYGQK, whereas the other four WRKY genes (SsWRKY14, SsWRKY23, SsWRKY38 and SsWRKY78) had a single mismatched amino acid in their core WRKY domains (Fig. 1). In SsWRKY14 and SsWRKY38, the WRKY domain has the sequence WRKYGKK, while SsWRKY23 contains a WKKYGQK sequence, and SsWRKY78 contains WRKYGRK sequence. Eulgem (2000) previously described that the zinc finger motif (C-X4–5-X22–23-H-X1-H or C-X7-C-X23-H-X1-C) is another vital feature of the WRKY family. As illustrated in Fig. 1, four WRKY domains (SsWRKY76C, SsWRKY64, SsWRKY12 and SsWRKY28) do not contain any distinct zinc finger motif, but they were still reserved in the succeeding analyses, as performed in barley and poplar (He et al., 2012; Mangelsen et al., 2008). Additionally, some zinc-finger-like motifs, including C-X4-C-X21-H-X1-H in SsWRKY23 and C-X5-C-X19-H-X1-H in SsWRKY73 and SsWRKY17, were identified in willow WRKY genes. Both the two zinc-finger-like motifs were also found in poplar (PtWRKY39, 57, 42 and 53).

Figure 1 Comparison of the WRKY domain sequences from 85 SsWRKY genes.

The WRKY gene with the suffix -N and -C indicates the N-terminal and C-terminal WRKY domain of group I members, respectively. “-” has been inserted for the optimal alignment. Red indicates the highly conserved WRKYGQK heptapeptide, and the zinc finger motifs are highlighted in green. The position of a conserved intron is indicated by an arrowhead.

Detailed characteristics of SsWRKY genes are listed in Table 1, including the WRKY gene specific group numbers, chromosomal distribution, Arabidopsis and poplar orthologs. The MW, PI and the length of each WRKY protein sequence are also shown in Table 1. According to the particularization (Table 1), the average length of these protein sequences is 407 residues, and the lengths ranged from 109 residues (SsWRKY23) to 1,593 residues (SsWRKY78). Additionally, the PI ranged from 5.03 (SsWRKY38, SsWRKY60) to 10.27 (SsWRKY28), and the MW ranged from 12.9 (SsWRKY23) to 179.0 kDa (SsWRKY78).

Table 1 The detailed characteristics of WRKY genes identified in willow.

Gene	Sequence ID	Chr	Group	Ortholog	Deduced polypeptide	Introns	
AtWRKY	PtWRKY	Length (aa)	PI	MW (kDa)	
SsWRKY1	willow_GLEAN_10011238	1	I	33	17	583	7.14	64.7	4	
SsWRKY2	willow_GLEAN_10019192	1	IIc	45	43	162	9.47	18.6	1	
SsWRKY3	willow_GLEAN_10017208	1	IIc	28,71	29	584	9.42	65.6	4	
SsWRKY4	willow_GLEAN_10017139	1	I	20	44	560	6.99	60.9	5	
SsWRKY5	willow_GLEAN_10007860	1	IIe	35	45	445	5.92	48.4	2	
SsWRKY6	willow_GLEAN_10003806	1	I	2	37,101,102	733	5.69	78.8	4	
SsWRKY7	willow_GLEAN_10022392	2	IId	21	46,63	453	9.53	49.9	4	
SsWRKY8	willow_GLEAN_10022273	2	IIc	71	47	328	6.89	37.0	2	
SsWRKY9	willow_GLEAN_10009329	2	IId	15	14,94	339	9.77	37.5	2	
SsWRKY10	willow_GLEAN_10009231	2	IIc	12	48	204	7.64	23.6	3	
SsWRKY11	willow_GLEAN_10016913	2	III	30	6,51	351	6.27	39.2	2	
SsWRKY12	willow_GLEAN_10016886	2	IIc	–	19,50	129	6.75	14.6	0	
SsWRKY13	willow_GLEAN_10016883	2	IIe	22	23,49,78	352	5.81	38.3	2	
SsWRKY14	willow_GLEAN_10019911	2	IIe	–	3	247	5.58	28.1	2	
SsWRKY15	willow_GLEAN_10019925	2	IIc	23	13,33	319	6.46	35.6	2	
SsWRKY16	willow_GLEAN_10019982	2	I	1	54	472	6.88	52.2	3	
SsWRKY17	willow_GLEAN_10020022	2	IIb	47	53	1,081	5.25	116.8	17	
SsWRKY18	willow_GLEAN_10025583	3	IId	–	55	142	9.60	16.5	2	
SsWRKY19	willow_GLEAN_10025423	3	IIe	29	41	335	5.54	37.9	2	
SsWRKY20	willow_GLEAN_10025378	3	III	41/53	21	342	5.25	38.4	2	
SsWRKY21	willow_GLEAN_10008020	3	IIc	45	18	157	9.41	17.8	1	
SsWRKY22	willow_GLEAN_10006448	3	IIa	40	88	320	8.38	35.4	3	
SsWRKY23	willow_GLEAN_10013342	3	IIc	–	39	109	8.03	12.9	1	
SsWRKY24	willow_GLEAN_10009960	4	IIb	42	28,79	604	6.93	65.3	5	
SsWRKY25	willow_GLEAN_10017267	4	IIe	65	8,58	267	5.43	29.7	2	
SsWRKY26	willow_GLEAN_10018559	4	I	58	60	537	8.72	58.9	3	
SsWRKY27	willow_GLEAN_10004854	4	III	54	85	323	5.70	36.3	2	
SsWRKY28	willow_GLEAN_10008312	5	IId	–	–	490	10.27	54.0	2	
SsWRKY29	willow_GLEAN_10009112	5	IIc	13	68	235	8.70	26.7	2	
SsWRKY30	willow_GLEAN_10003565	5	IId	15	20	310	9.48	34.3	2	
SsWRKY31	willow_GLEAN_10016009	5	IIc	28,71	62	322	6.67	36.2	2	
SsWRKY32	willow_GLEAN_10018195	5	IId	21	46,63	349	9.69	38.8	2	
SsWRKY33	willow_GLEAN_10026833	6	IId	7	91	339	9.89	36.8	3	
SsWRKY34	willow_GLEAN_10026721	6	IIc	49	34	287	5.25	32.1	2	
SsWRKY35	willow_GLEAN_10026591	6	I	33	64	572	6.41	62.7	4	
SsWRKY36	willow_GLEAN_10026566	6	III	54	85	329	6.13	36.7	2	
SsWRKY37	willow_GLEAN_10020588	6	I	44	93	478	9.25	52.5	4	
SsWRKY38	willow_GLEAN_10026166	6	IIc	51	67	233	5.03	26.1	2	
SsWRKY39	willow_GLEAN_10026455	6	IIa	18/60	9	327	9.02	36.2	4	
SsWRKY40	willow_GLEAN_10026458	6	I	32	15	413	8.26	44.9	3	
SsWRKY41	willow_GLEAN_10008192	7	IIc	13	68	236	9.21	26.6	2	
SsWRKY42	willow_GLEAN_10025108	8	I	3/4	69	460	8.80	50.6	3	
SsWRKY43	willow_GLEAN_10025123	8	IIc	57	71	295	6.32	32.3	2	
SsWRKY44	willow_GLEAN_10015641	8	IIc	48	70	357	6.11	39.9	2	
SsWRKY45	willow_GLEAN_10008155	9	IId	15	20,26	331	9.57	36.4	2	
SsWRKY46	willow_GLEAN_10013562	10	IIc	57	71	289	6.26	31.9	2	
SsWRKY47	willow_GLEAN_10013586	10	I	3/4	72	490	8.60	53.7	3	
SsWRKY48	willow_GLEAN_10004012	11	IIb	42	100	585	6.48	63.3	5	
SsWRKY49	willow_GLEAN_10006060	11	I	20	44	607	7.09	6.6	6	
SsWRKY50	willow_GLEAN_10007614	11	IIe	35	74	481	5.39	51.6	3	
SsWRKY51	willow_GLEAN_10007542	11	I	2	37	734	6.10	79.7	4	
SsWRKY52	willow_GLEAN_10013801	12	IIc	–	75	178	9.08	20.5	1	
SsWRKY53	willow_GLEAN_10012158	13	IId	74	25	356	9.66	40.0	2	
SsWRKY54	willow_GLEAN_10004417	13	I	2	35	697	6.52	76.1	4	
SsWRKY55	willow_GLEAN_10007732	13	I	33	1	602	7.65	66.0	4	
SsWRKY56	willow_GLEAN_10009039	14	IId	15	14,94	362	9.39	40.0	2	
SsWRKY57	willow_GLEAN_10016668	14	IIc	12	48	180	8.47	20.7	3	
SsWRKY58	willow_GLEAN_10016177	14	IIe	22	23,49,78	354	6.35	38.8	2	
SsWRKY59	willow_GLEAN_10016180	14	IIc	43	19,50	193	9.47	21.7	1	
SsWRKY60	willow_GLEAN_10016220	14	III	30	6	368	5.03	41.3	2	
SsWRKY61	willow_GLEAN_10018940	14	IIb	42	28,79	467	8.78	50.0	5	
SsWRKY62	willow_GLEAN_10018891	14	IIc	23	13,33	318	5.71	35.6	2	
SsWRKY63	willow_GLEAN_10018881	14	IIe	–	80	263	5.05	29.7	2	
SsWRKY64	willow_GLEAN_10020302	14	IIb	36	–	460	6.28	50.0	4	
SsWRKY65	willow_GLEAN_10020380	14	I	1	2	481	5.98	52.8	3	
SsWRKY66	willow_GLEAN_10011119	15	IIb	9	99	618	6.55	66.2	5	
SsWRKY67	willow_GLEAN_10016438	15	IIc	–	82	178	9.35	20.5	1	
SsWRKY68	willow_GLEAN_10023347	16	IIa	40	88	320	8.82	35.3	3	
SsWRKY69	willow_GLEAN_10023447	16	IIc	45	18	178	9.17	20.1	1	
SsWRKY70	willow_GLEAN_10023687	16	III	41/53	21	336	5.17	37.2	2	
SsWRKY71	willow_GLEAN_10023735	16	IIe	29	41	325	5.54	36.6	2	
SsWRKY72	willow_GLEAN_10014752	16	IId	–	55	338	9.24	37.9	2	
SsWRKY73	willow_GLEAN_10009602	16	IIb	9	42	509	5.51	55.3	4	
SsWRKY74	willow_GLEAN_10010473	17	IIc	45	43	182	9.92	20.9	1	
SsWRKY75	willow_GLEAN_10015128	17	IIb	9	86	544	6.01	59.0	3	
SsWRKY76	willow_GLEAN_10015184	17	I	58	87	1,044	8.94	116.1	11	
SsWRKY77	willow_GLEAN_10005468	17	IIe	27	96	411	5.96	45.7	2	
SsWRKY78	willow_GLEAN_10006860	18	I	–	90	1,593	8.67	179.0	10	
SsWRKY79	willow_GLEAN_10006862	18	IIa	18/60	9	320	8.57	35.6	4	
SsWRKY80	willow_GLEAN_10011608	18	I	32	–	528	5.74	57.8	4	
SsWRKY81	willow_GLEAN_10004546	18	IId	7	7,91	300	9.80	32.8	2	
SsWRKY82	willow_GLEAN_10003422	19	IId	11/17	24	339	9.58	37.1	2	
SsWRKY83	willow_GLEAN_10011321	19	III	55	36,76	358	5.63	38.7	2	
SsWRKY84	willow_GLEAN_10005288	19	I	33	4	597	6.69	65.6	4	
SsWRKY85	willow_GLEAN_10002834	N/A	IIe	65	58	268	5.83	30.2	2	
Notes:

Chr, chromosome numbers.

N/A, not available.

“–”, not detected.

Locations and gene clusters of willow WRKY genes

Nearly 84 of the 85 putative SsWRKY genes could be mapped onto 19 willow chromosomes and then renamed from SsWRKY1 to SsWRKY84 based on their specific distributions on chromosomes. Only one SsWRKY gene (willow_GLEAN_10002834), renamed as SsWRKY85, could not be conclusively mapped onto any chromosome. As shown in Fig. 2, Chromosome (Chr) 2 possessed the largest number of SsWRKY genes (11 genes), followed by Chr14 (10 genes). Eight SsWRKY genes were found on Chr6, six on Chr1 and Chr16, and five on Chr5. Additionally, four chromosomes (Chr4, Chr11, Chr17, Chr18) had four SsWRKY genes, as well as three SsWRKY genes were found on Chr8, Chr13 and Chr19. Chr10 and Chr15 had two SsWRKY genes, and only one SsWRKY gene was identified on Chr7, Chr9 and Chr12. The distribution of each SsWRKY genes was extremely irregular, indicating the reduction of the TDs in willow WRKY genes.

Figure 2 Chromosomal location of SsWRKY genes.

Red indicates group I, blue indicates group II and green indicates group III. Red lines indicate gene clusters. The chromosome numbers are given at the top of each chromosome and the left side of each chromosome is related to the approximate physical location of each WRKY gene. Only one unmapped SsWRKY gene is shown on ChrN.

As described by Holub (2001), a single chromosome region containing two or more genes within 200 kb was defined as gene clusters (He et al., 2012). According to this description, a total of 23 SsWRKY genes were clustered into 11 clusters in willow (Fig. 2). The chromosomal distribution of gene cluster was irregular, and only seven chromosomes were identified to have gene clusters. Three clusters, including seven SsWRKY genes, were found on Chr2, and two clusters were found on both Chr6 and Chr14. Only one cluster was distributed on each of Chr3, Chr8, Chr10 and Chr18, whereas none was identified on other 11 chromosomes. Further analysis of SsWRKY chromosomal distribution showed that a high WRKY gene density region in only 2.23 Mb regions on Chr2, which had also been observed in rice and poplar (He et al., 2012; Wu, 2005).

Phylogenetic analysis and classification of WRKY genes in willow

In order to get a better separation of different groups and subgroups in SsWRKY genes, a total of 185 WRKY domains, including 82 AtWRKY domains and 103 SsWRKY domains, were used to construct the NJ phylogenetic tree. On the basis of the phylogenetic tree and structural features of WRKY domains, all 85 SsWRKY genes were clustered into three main groups (Fig. 3). Nineteen members containing two WRKY domains and C2H2-type zinc finger motifs were categorized into group I, except SsWRKY78, which contains only one WRKY domain and two zinc finger motifs. Domain acquisition and loss events appear to have shaped the WRKY family (Ross, Liu & Shen, 2007; Rossberg et al., 2001). Thus, SsWRKY78 may have evolved from a two-domain WRKY gene but lost one WRKY domain during evolution. Additionally, as shown in Fig. 3, SsWRKY78 shows high similarities to SsWRKY40N, implying a common origin of their domains. The similar phenomenon was also found in PtWRKY90 of poplar (He et al., 2012).

Figure 3 Phylogenetic tree of WRKY domains from willow and Arabidopsis.

The phylogenetic tree was constructed using the neighbor-joining method in MEGA 6.0. The WRKY genes with the suffix ‘N’ and ‘C’ indicate the N-terminal and the C-terminal WRKY domains of group I, respectively. The different colors indicate different groups (I, II and III) or subgroups (IIa, b, c, d and e) of WRKY domains. Circles indicate WRKY genes from willow, and diamonds represent genes from Arabidopsis. The purple trapezoid region indicate a new subgroup belonging to IIc.

The largest number of SsWRKY genes, comprising a single WRKY domain and C2H2 zinc finger motif, were categorized into group II. SsWRKY genes of group II could be further divided into five subgroups: IIa, IIb, IIc, IId and IIe. As shown in Fig. 3, subgroup IIa (four members) and IIb (eight members) were clustered into one clade, as well as subgroup IId (13 members) and IIe (11 members). Strikingly, SsWRKY genes in subgroup IIc (21 members) and group IC are classified into one clade, suggesting that group II genes are not monophyletic and the group IIc WRKY genes may evolve from the group I genes by the loss of the WRKY domain in N-terminal. As shown in Figs. 3 and S1, SsWRKY23, SsWRKY34 and their orthologous genes (AtWRKY49, PtWRKY39, PtWRKY57, PtWRKY34 and PtWRKY32) seem to form a new subgroup closer to the group III. However, SsWRKY23 and SsWRKY34 exhibit the zinc finger motif C-X4-C-X21-H-X-H and C-X4-C-X23-H-X-H as observed in the subgroup IIc and group IC. Therefore, they were classified into subgroup IIc in this study.

Different from the C2H2 zinc finger pattern in group I and II, group III WRKY genes (seven members), broadly considered as playing vital roles in plant evolution process and adaptability, contained one WRKY domain and a C-X7-C-X23-H-X-C zinc finger motif. However, in rice and barley, a new CX7CXnHX1C (n ≥ 24) zinc finger motif was identified in group III (Mangelsen et al., 2008; Wu, 2005), which was never found in poplar, grape, Arabidopsis and willow, suggesting that this feature perhaps only belong to monocotyledonous species.

In order to obtain a better study in woody plant species, a phylogenetic tree based on the WRKY domains between willow and poplar was constructed (Fig. S1). The tree showed that most of the WRKY domains from willow and poplar were clustered into sister pairs, suggesting that gene duplication events played prominent roles in the evolution and expansion of WRKY gene family. Furthermore, a total of 20 SsWRKY domains show extremely the same domains (similarity: 100%) to poplar, i.e., SsWRKY39 and PtWRKY9, SsWRKY39 and PtWRKY9, SsWRKY39 and PtWRKY9, SsWRKY39 and PtWRKY9, and so on. Further functional analyses of these genes in willow or poplar will provide a useful reference for another one.

The ortholog of SsWRKY genes in Arabidopsis and poplar

The clustering of orthologous genes emphasizes the conservation and divergence of gene families, and they may contain the same functions (Ling et al., 2011). In this study, a phylogeny-based method was used to identify the putative orthologous SsWRKY genes in Arabidopsis and poplar (Figs. 3 and S1), and BLAST-based method (Bi-direction best hit) was used to confirm the true orthologs. The WRKY genes of group I contained two WRKY domains, and both of them were used to construct the phylogenetic trees. To avoid the mistakes of orthologous genes in group I, the members of group I WRKY genes were considered as orthologous genes unless the same phylogenetic relationship can be detected between N-termini and C-termini in the phylogenetic tree. For example, SsWRKY37 and AtWRKY44 were considered as an orthologous gene pair because they clustered into a clade of their N-termini and C-termini (Fig. 3), while SsWRKY80 and PtWRKY30 were excluded from orthologous gene pairs due to their different clusters of N-termini and C-termini (Fig. S1). Totally, 75 orthologous gene pairs were found between willow and Arabidopsis, less than 82 orthologous gene pairs between willow and poplar (Table 1), which was congruent with the evolutionary relationship among the three plant species.

Evolutionary analysis of WRKY III genes in willow

The WRKY III genes were considered as the evolutionary youngest groups, and played crucial roles in the process of plant growth and resistance. In order to further probe the duplication and diversification of WRKY III genes after the divergence of the monocots and dicots, a phylogenetic tree was constructed using 65 WRKY III genes from Arabidopsis (13), rice (29), poplar (10), willow (7) and grape (6). As shown in Fig. S2, willow SsWRKY III genes were closer to the eurosids I group (poplar and grape) than eurosids II group (Arabidopsis) and monocots (rice). Meanwhile, most Arabidopsis and rice WRKY III genes formed the relatively independent clades, suggesting that two gene duplication events, including tandem and segmental duplication, perhaps were the main factors in the expansion of WRKY III genes in Arabidopsis and rice. The results also indicated that WRKY III genes might arise after the divergence of the Arabidopsis (eurosids I) and eurosids II (poplar, willow and grape). The study by Ling et al. (2011) in cucumber showed the similar results and hence proved the validity. Additionally, we found that seven rice WRKY III genes (OsWRKY55, 84, 18, 52, 46, 114 and 97) contained the variant domain WRKYGEK, which was not found in other four dicots (Arabidopsis, poplar, grape and willow), implying that this may be a feature of WRKY III genes in monocots and these OsWRKY genes may respond to different environmental signals.

According to the comparison of the number of WRKY III genes in the five observed plants, the number is smaller in eurosids I (poplar, grape and willow) than Arabidopsis (eurosids II) and rice (monocots), which may be caused by different patterns of duplication events. Genes generated by duplication events are not stable, and can be retained or lost due to different selection pressure and evolution (Zhang, 2003). In order to determine which selection pressure played prominent roles in the expansion of willow WRKY III genes, we estimated the Ka/Ks ratios for all pairs (21 pairs) of willow WRKY III genes. As shown in Table S1, all the Ka/Ks ratios were less than 0.5, suggesting willow WRKY III genes had mainly been subjected to strong purifying selection and they were slowing evolving at the protein level.

Exon–intron structures of SsWRKY genes

The exon–intron structures of multiple gene families play crucial roles during plant evolution. As shown in Fig. 4, the SsWRKY gene phylogenetic tree and the corresponding exon–intron structures are shown in A and B, respectively. Exon–intron structures of each group were shown in Fig. 4B, a large number of WRKY genes had two to five introns (94%, 80 of 85), including eight WRKY genes contained one intron; 39 contained two introns; 13 contained three introns; 15 contained four introns and 5 contained five introns. The number of exons in remaining WRKY genes was quite different: SsWRKY49, SsWRKY76 and SsWRKY78 had 6, 11 and 10 introns, respectively; SsWRKY17 had the largest number of introns (17 introns), while no intron was found in SsWRKY12. The intron acquisition or loss occurred during the evolution of WRKY gene family, while WRKY genes in the same group shared the similar number of introns (Guo et al., 2014). In our study, most of WRKY genes in group I had three to six introns, expect SsWRKY76 and SsWRKY78, which might acquire some introns during evolution. The number of introns of WRKY genes in group II was extremely different, ranging from one to five introns, except SsWRKY17 with 17 introns and SsWRKY12 with zero intron might obtain or loss some introns during evolution. Strikingly, WRKY genes in group III had the most stable number of introns with all of seven WRKY III genes had two introns, suggesting that WRKY III genes may be the most stable genes in the environmental stress. The stable number of introns in SsWRKY III genes was consistent with the results of Ka/Ks analysis, which reflected that purifying selection pressure played vital roles in willow WRKY III genes.

Figure 4 Genomic organization of SsWRKY genes.

(A) The phylogenetic tree built on the basis of full-length SsWRKY genes was depicted using the neighbor-joining method in MEGA 6.0. The short black lines indicate the existence of duplicated gene pairs; (B) The graphic exon–intron structure of SsWRKY genes is displayed using GSDS. Green indicates exons, and gray indicates introns. The introns phases 0, 1 and 2 are indicated by numbers 0, 1 and 2, respectively.

A great deal of studies in WRKY genes proved that nearly all of the WRKY genes contained an intron in their WRKY core domains (Eulgem, 2000; Guo et al., 2014; He et al., 2012; Huang et al., 2012; Ling et al., 2011; Zou et al., 2004). According to the further analysis of SsWRKY genes, two major types of splicing introns, R-type and V-type, introns were observed in numerous SsWRKY domains. The R-type intron was spliced exactly at the R residue, about five amino acids before the first Cys residue in the C2H2 zinc finger motif. The V-type intron was localized before the V residue, six amino acids after the second Cys residue in the C2H2 zinc finger motif. As shown in Fig. 4B, the R-type introns could be observed in more groups, including group IC, subgroup IIc, IId, IIe and group III, while V-type introns were only observed in subgroup IIa and IIb. However, there was no intron found in group IN. The similar results were also observed in Arabidopsis, poplar and rice, suggesting that the special distribution of introns in WRKY domains was a feature of WRKY family (Eulgem, 2000; He et al., 2012; Wu, 2005).

Identification of gene duplication events and conserved motifs in willow

Gene duplication events were always considered as the vital sources of biological evolution (Chothia et al., 2003; Ohno, Wolf & Atkin, 1968). TDs were defined as two or more adjacent homologous genes located on a single chromosome, while homologous gene pairs between different chromosomes were defined as SDs (Liu & Ekramoddoullah, 2009). In our study, a total of 33 homologous gene pairs, including 66 SsWRKY genes, were identified to participate in gene duplication events (Table S2). The composition of gene duplication events in each group in ascending order was group I: 73.7% (14 of 19), group II: 78% (46 of 59) and group III: 85.7% (6 of 7). Among the 33 homologous gene pairs, none of them appeared to have undergone TDs, on the contrary, all of the 66 genes (77.6% of all SsWRKY genes) participated in SDs, implying that SDs played major roles in the expansion of willow WRKY genes.

WRKY genes shared more functional and homologies in their conserved WRKY core domains (about 60 residues), while the rest sequences of WRKY genes shared a little (Eulgem, 2000). In order to get a more comprehensive understanding of the structural feature in WRKY domains, the conserved motifs of SsWRKY genes were predicted using the online program MEME (Fig. S3; Table S3). Among the 20 putative motifs, motifs 1, 2, 3 and 5, broadly distributed across SsWRKY genes, were characterized as the WRKY conserved domains. The motif 6 was characterized as nuclear localization signals (NLS), which mainly distributed in subgroup IId and IIe and group III. Some other motifs with poorly defined recently were also predicted by MEME: the motif 4 was only found in group IC and subgroup IIc; motifs 7 and 9 were limited to subgroup IIa and IIb; the motif 8 was found in group I and a few genes of subgroup IIc; motifs 10, 13, 15 and 17 were unique in subgroup IId; the motif 12 was only observed in subgroup IIb; the motif 16 was mainly found in group II; the motif 18 was found in subgroup IIc; motifs 19 and 20 were only observed in subgroup I. The distinct conserved motifs of different groups could be an important foundation for future structural and functional study in WRKY gene family.

Some other important motifs, including Leu zipper motif, HARF, LXXLL and LXLXLX, could be also identified in WRKY genes. Using the online program 2ZIP, the conserved Leu zipper motif, described as a common hypothetical structure to DNA binding proteins (McInerney et al., 1998), was identified in only two SsWRKY genes (SsWRKY61 and SsWRKY39). With manual inspection, the conserved HARF (RTGHARFRR[A/G]P) motifs, whose putative functions were not distinguished clearly, were only observed in seven WRKY genes of subgroup IId, including SsWRKY82, 33, 45, 81, 9, 30 and 56. In the meantime, the conserved LXXLL and LXLXLX (L: Leucine; X: any amino acid) motifs, which respectively defined as the co-activator and active repressor motifs, were also found in SsWRKY genes. A total of seven SsWRKY genes (SsWRKY19, 45, 72, 61, 76, 30 and 59) contained the helical motif LXXLL, whereas eight genes (SsWRKY66, 26, 35, 81, 83, 75, 73 and 3) shared the LXLXLX motif. The plenty of conserved motifs in WRKY genes with different lengths and variant functions, suggesting that the WRKY genes might play more vital roles in gene regulatory network.

Distinct expression profiles of SsWRKY genes in various tissues

In order to gain more information about the roles of WRKY genes in willow, RNA-seq data from the sequenced genotype were used to quantify the expression level of WRKY genes in five tissues of Salix suchowensis. As illustrated in Fig. 5, the expression of all 85 SsWRKY genes were detected in at least one of the five examined tissues, such as 84 genes in roots, 80 in stems, 84 in barks, all in buds and 73 in leaves. Meanwhile, the cluster analysis of the expression pattern in five tissues showed that SsWRKY genes shared more similarities between stem and leaf, as well as bark and bud, and root was more similar to the clade formed by bark and bud. The results detected here were consistent with their biological characteristics. SsWRKY38, not detected in roots and leaves, was also lowly expressed in other tissues. Similarly, SsWRKY74, not detected in stems, barks and leaves, was only expressed in roots and buds with extremely low levels. Among the five genes not expressed in stems, SsWRKY66, 74 and 79 were also not detected in leaves. The largest number of expressed or unexpressed SsWRKY genes (12 genes) was found in buds or leaves, respectively, suggesting that WRKY genes might play more roles in buds than leaves.

Figure 5 Expression profiles of the 85 SsWRKY genes in root, stem, bark, bud and leaf.

Color scale represents RPKM normalized log2 transformed counts and red indicates high expression, blue indicates low expression and white indicates the gene is not expressed in this tissue.

According to the expression annotation of 85 SsWRKY genes by RPKM method in Fig. 5 and Table S4, the total transcript abundance of SsWRKY genes in tender root (RPKM = 1,181.21), bark (RPKM = 1,363.01) and vegetative bud (RPKM = 928.58) was relatively larger than that in other two tissues, including non-lignified stem (RPKM = 537.88) and young leaf (RPKM = 349.84). As shown in Table S4, SsWRKY81 (RPKM = 97.75), the most expressed SsWRKY genes in roots, was also expressed in other four tissues, though the expression levels were relatively low; SsWRKY56 (RPKM = 32.54), the most expressed SsWRKY genes in stem, was also highly expressed in other examined tissues. Similarly, SsWRKY67, the most expressed SsWRKY genes in barks (RPKM = 188.16), was also detected in vegetative buds (RPKM = 82.07) and young leaves (RPKM = 26.11) with high expression levels. Similarly, SsWRKY6 (RPKM = 26.31), the most expressed genes in leaves, was also highly detected in other tissues. A few genes, i.e., SsWRKY52, SsWRKY2 and SsWRKY35, were expressed highly in barks, but lowly in other four tissues. The results mentioned above may be an important foundation for the specific expression analysis of each WRKY gene in willow.

Discussion

The WRKY transcription factor gene family can specifically interact with the W-box to regulate the expressions of downstream target genes. They also play prominent roles in diverse physiological and growing processes, especially in various abiotic and biotic stress responses in plants. Previous studies about the features and functions of WRKY family have been conducted in many model plants, including Arabidopsis for annual herbaceous dicots (Eulgem, 2000), grape for perennial dicots (Guo et al., 2014), poplar for woody plants and rice for monocots (He et al., 2012; Wu, 2005), but there is no large-scale study of WRKY genes in willow. Here, the comprehensive analysis of WRKY family in willow (Salix suchowensis) would facilitate a better understanding of WRKY gene superfamily and provide interesting gene pools to be investigated for breeding and genetic engineering purposes in woody plants.

As described in many previous studies, the presence of highly conserved WRKY domains in WRKY proteins is the most prominent characteristic of the WRKY gene family (Ding et al., 2015; Eulgem, 2000; He et al., 2012; Huang et al., 2012; Wu, 2005). In our study, through comparing the two phylogenetic trees based on the conserved WRKY domains (Fig. 3) and proteins (Fig. 4A), we obtained the nearly same classification of all SsWRKY genes, suggesting that the conserved WRKY domain is an indispensable unit in WRKY genes. The variation of the WRKYGQK heptapeptide may influence the proper DNA-binging ability of WRKY genes (Duan et al., 2007; Maeo et al., 2001). A recent binding study by Brand et al. (2013) disclosed that a reciprocal Q/K change of the WRKYGQK heptapeptide might result in different DNA-binding specificities of the respective WRKY genes. For instance, the soybean WRKY genes, GmWRKY6 and GmWRKY21, which contains the WRKYGKK variant, can’t bind normally to the W-box (Zhou et al., 2008). NtWRKY12 gene in tobacco with the WRKYGKK variant recognizes another binding sequence ‘TTTTCCAC’ instead of normal W-box (van Verk et al., 2008). In our study, four WRKY genes (SsWRKY14, SsWRKY23, SsWRKY38 and SsWRKY78) had a single mismatched amino acid in their conserved WRKYGQK heptapeptide (Fig. 1). The variants detected in willow were extremely congruent with that in another salicaceous plant, poplar, which also contains the same three variants in seven PtWRKY genes (He et al., 2012). Previous studies have disclosed that the binding specificities of variable WRKYGQK heptapeptide vary tremendously (Brand et al., 2013); however, few studies were shown about the effect of variable zinc finger motif. In this study, four WRKY domains (SsWRKY76C, SsWRKY64, SsWRKY12 and SsWRKY28) without complete zinc finger motif may lack the ability of interacting with W-box, as well as PtWRKY83, 40, 95 and 10 in poplar (He et al., 2012). Thereby, it is still indispensable to further investigate the function or the expression patterns of the regulated gene targets in the variant sequences of the WRKY domains (both WRKYGQK heptapeptide and complete zinc finger motif).

Different classification methods may lead to different numbers of WRKY genes in each group. The classification method in our study was categorized as described in Arabidopsis, grape, cucumber, castor bean and many other plant species (Eulgem, 2000; Guo et al., 2014; Ling et al., 2011; Zou et al., 2016). According to this method, the willow WRKY genes were classified into three main groups (I, II and III), with five subgroups in group II (IIa, IIb, IIc, IId and IIe). However, the strategy described in rice and poplar was a little different (He et al., 2012; Wu, 2005). They classified the subgroup IIc categorized above into a new subgroup Ib based on the fact that the C-termini of group I and the domains of the above subgroup IIc shared more similar consensus structures. At the meantime, subgroup IId and IIe categorized above were reclassified into subgroup IIc and IId, respectively. With the same classification method as described in Arabidopsis and many other plants, the numbers of different groups in poplar and rice are illustrated in Table S5. WRKY genes of subgroup IIa, the smallest number of members, appear to play crucial roles in regulating biotic and abiotic stress responses (Rushton et al., 2010). As shown in Table S5, the willow WRKY genes of subgroup IIa and IIb are extremely similar to that of other plant species, suggesting that all SsWRKY genes of these subgroups have been identified. In addition, the numbers of WRKY III in eurosids I group, such as cucumber (6), poplar (10), grape (6) and willow (7) are less than that of eurosids II (Arabidopsis: 14) and monocots (rice: 36), suggesting that different duplication events or selection pressures occurred in WRKY III genes after the divergence of eurosids I and eurosids II group. A previous study in Arabidopsis showed that nearly all WRKY III members respond to diverse biotic stresses, indicating that this group probably evolved with the increasing biological requirements (Wang et al., 2015). The different numbers of WRKY III genes in willow, poplar, cucumber, Arabidopsis and rice are probably due to their different biotic stresses during evolution, and seven SsWRKY III genes may be sufficient for the biological requirements in willow.

WRKY transcription factors play important roles in the regulation of developmental processes and response to biotic and abiotic stress (Brand et al., 2013). The evolutionary relationship of WRKY gene family promises to obtain significant insights into how biotic and abiotic stress responses from single cellular aquatic algae to multicellular flowering plants (Rinerson et al., 2015). Previous studies hypothesized that group I WRKY genes were generated by domain duplication of a proto-WRKY gene with a single WRKY domain, group II WRKY genes evolved through the subsequent loss of N-terminal WRKY domain, and group III genes evolved from the replacement of conserved His residue with a Cys residue in zinc motif (Wu, 2005). However, recent study proposed two alternative hypotheses of WRKY gene evolution (Rinerson et al., 2015): the “Group I Hypothesis” and the “IIa + b Separate Hypothesis.” Additionally, another recent study by Brand et al. (2013) concluded that subgroup IIc WRKY genes evolved directly from IIc-like ancestral WRKY domains, and group I genes evolved independently due to a duplication of the IIc-like ancestral WRKY domains. Phylogenetic analysis in our study shows that subgroup IIc and group IC are evolutionarily close, as well as subgroups IIa and IIb, subgroups IId and IIe, and this result is consistent with the conclusion drew by Brand et al. (2013). Additionally, the V-type introns of SsWRKY genes are only found in subgroup IIa and IIb, while R-type introns are found in other groups except group IN. The results are congruent with the “IIa + b Separate Hypothesis.” Our results shown here provide important reference for the further analyses on the accurate evolutionary relationship of WRKY gene family.

Gene duplication events played prominent roles in a succession of genomic rearrangements and expansions, and it is also the main motivation of plants evolution (Vision, Brown & Tanksley, 2000). The gene family expansion occurs via three mechanisms: TDs, SDs and transposition events (Maher, Stein & Ware, 2006), and we only focused on the TDs and SDs in this study. In willow, a total of 66 SsWRKY genes were identified to participate in gene duplication events, and all of these genes appeared to have undergone SDs. Similarly, in poplar, only one homologous gene pair participated in TDs, while 29 of 42 (69%) homologous gene pairs were determined to participate in SDs. The similar WRKY gene expansion patterns in willow and poplar showed that SDs were the main factors in the expansion of WRKY genes in woody plants. However, in cucumber, no gene duplication events have occurred in CsWRKY gene evolution, probably because there were no recent whole-genome duplication and tandem duplication in cucumber genome (Huang et al., 2009). In rice and Arabidopsis, many WRKY genes were generated by TDs, which was incongruent with the duplication events in willow, poplar and cucumber. The different WRKY gene expansion patterns of the above plant species could be due to their different life habits and selection pressures in a large scale, and it is still indispensable to be further investigated.

The WRKY gene family plays crucial roles in response to biotic and abiotic stresses, as well as diverse physiological and developmental processes in plant species. Because of the lack of researches on the function of willow WRKY genes, our study provided putative functions of SsWRKY genes by comparing the orthologous genes between willow and Arabidopsis. The details of the functions or regulations of AtWRKY genes can be obtained from TAIR (http://www.arabidopsis.org/). For example, AtWRKY2, the ortholog to SsWRKY6, which highly expressed in the five examined tissues, plays important roles in seed germination and post germination growth (Jiang & Yu, 2009). AtWRKY33, the ortholog to SsWRKY1, 35, 55 and 84, influences the tolerance to NaCl, and increases sensitivity to oxidative stress and abscisic acid (Jiang & Deyholos, 2009). A large number of AtWRKY genes, i.e. AtWRKY3, 4, 18, 53, 41, work in the resistance to Pseudomonas syringae (Chen & Chen, 2002; Higashi et al., 2008; Lai et al., 2008; Murray et al., 2007), therefore their orthologs in willow (SsWRKY42, 47, 39, 79, 20 and 70) may show the same resistance to Pseudomonas syringae. Based on the comparison of willow WRKY genes with their Arabidopsis orthologs, we could speculate that the functional divergence of SsWRKY genes has played prominent roles in the responses to various stresses.

Conclusions

Based on the recent released willow genome sequence and RNA-seq data, in this study we identified 85 SsWRKY proteins using bioinformatics approach. According to the phylogenetic relationships and structural features of WRKY domains, all 85 SsWRKY genes were assigned to the group I, group II (subgroup a–e) and group III. Three variations of the WRKYGQK heptapeptide and the normal zinc finger motif in willow WRKY genes might execute some new biological functions. Evolutionary analysis of SsWRKY III genes will be helpful for understanding the evolution of WRKY III genes in plant. With the comparison of willow WRKY genes with their Arabidopsis orthologs, breeding willow varieties with increased tolerance to many adverse environments could be achieved using transgenic technology. Our results will be not only beneficial to complete the functional and annotation information of WRKY genes family in woody plants, but also provide interesting gene pools to be investigated for breeding and genetic engineering purposes in woody plants.

Supplemental Information

Supplemental Information 1 Phylogenetic tree of WRKY domains from willow and poplar.

The phylogenetic tree was constructed using the neighbor-joining method in MEGA 6.0. The WRKY genes with the suffix ‘N’ and ‘C’ indicate the N-terminal and the C-terminal WRKY domains of group I, respectively. The different colors indicate different groups (I, II and III) or subgroups (IIa, b, c, d and e) of WRKY domains. Circles indicate WRKY genes from willow, and triangles represent genes from poplar. The purple trapezoid region indicate a new subgroup belonging to IIc.

Click here for additional data file.

Supplemental Information 2 Phylogenetic tree of full-length group III WRKY genes from Arabidopsis (AtWRKY), rice (OsWRKY), grape (VvWRKY), poplar (PtWRKY) and willow (SsWRKY).

The phylogenetic tree was constructed using the neighbor-joining method in MEGA 6.0. Dicotyledonous (Arabidopsis, grape, poplar and willow) and monocotyledonous (rice) WRKY III genes are marked with colored dots.

Click here for additional data file.

Supplemental Information 3 The distribution of twenty conserved motifs of SsWRKY genes.

The names of all members are displayed on the left side of the figure. Different motifs are displayed in different colored boxes as indicated on the right side. The conserved motifs 1, 2, 3, and 5, broadly distributed across SsWRKY genes, were definitely characterized as the WRKY conserved domains.

Click here for additional data file.

Supplemental Information 4 The Ka/Ks ratios of 21 WRKY III gene pairs in Salix suchowensis.

Click here for additional data file.

Supplemental Information 5 The details of twenty conserved motif sequences identified in SsWRKY genes.

Click here for additional data file.

Supplemental Information 6 The details of 33 homologus SsWRKY gene pairs in Salix suchowensis.

Click here for additional data file.

Supplemental Information 7 The details of total transcript abundance of SsWRKY genes by RPKM annotation.

Click here for additional data file.

Supplemental Information 8 The number of WRKY genes identified in Arabidopsis thaliana, Cucumis sativus, Poplulus trichocarpa, Vitis vinifera, Salix suchowensis and Oryza sativa.

Click here for additional data file.

The authors are deeply grateful to Prof. Tongming Yin, who provided the draft willow genome and RNA-seq data from five tissues. The authors also thank Prof. Ning Ye, Prof. Qiaolin Ye and Dr. Yiqing Xu for providing valuable suggestions and comments.

Additional Information and Declarations

Competing Interests

Author Contributions

Data Deposition

The authors declare that they have no competing interests.

Changwei Bi conceived and designed the experiments, performed the experiments, analyzed the data, wrote the paper, prepared figures and/or tables.

Yiqing Xu analyzed the data, prepared figures and/or tables, reviewed drafts of the paper.

Qiaolin Ye wrote the paper.

Tongming Yin conceived and designed the experiments, contributed reagents/materials/analysis tools.

Ning Ye performed the experiments, reviewed drafts of the paper.

The following information was supplied regarding data availability:

The latest S. suchowensis genome (version5_2.fa), annotation information (version5_2.gff3), coding sequences (Willow.gene.cds) and protein sequences (Willow.gene.pep) are available at our laboratory website (http://bio.njfu.edu.cn/ss_wrky/).

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
