# Peer review of "Genome-wide identification and characterization of WRKY gene family in Salix suchowensis"

_PeerJ, doi:10.7717/peerj.2437_

## Round 0.1 · original submission · Major Revisions

Dear Authors

Your manuscript has been reviewed by two experts. Both of the reviewers were supportive on the significance of your findings. However, both of the reviewers also raised several major concerns on your manuscript. Especially, the manuscript have problems of redundancy and lack of describing implication of your findings.
I recommend English editing, major revision of your manuscrpt before resubmit for further round of review processes.

Thanks for your cooperations.

Reviewer 1 ·

Basic reporting

Major concers
1. The manuscript reads like a list of results, so there needs to be more discussion, and implication of the analysis. Much of discussion has redundancy and it feels like a review of other species WRKY rather than discussion of willow WRKY.

Minor concerns
1. Page 1, line 12; there are WRKY genes in the lineage of non-plant eukaryotes, as the authors mentioned.
2. Page 2, line 21; reference of cotton WRKY is missing (Ref32).
3. Page 3, line17-18; references are outdated.
4. Page 4, line6-11; the order of the sentences should be changed.
5. Page 9, line3-5; the list of excluded gene is not needed.
6. Page 13, line16-19; interesting point should be found in SsWRKY, not OsWRKY in this manuscript.
7. Page 15, line8; references are missing.
8. It might be better if the authors mark the WRKY domain region at figure 8 and table 2.
9. Page 18, line2; “the induced plant TFs” is too assertive.
10. The manuscript contains many minor errors in grammar. For example,
A. Page 3, line 1; “cis” should be italic.
B. Page 4, line 25; replace “higher” to “high”
C. Page 7, line4; “e-value < 1e-20” or “e-value cutoff = 1e-20
D. Page 9 line 26; replace “list” to “listed”
E. Page 12, line 1; “IIIb” does not exist.
11. References should be confirmed if they are appropriate at the sentence. For example,
A. Page 3, line 22; Ref3.
B. Page 4, line 7; Ref36
C. Page 4, line10; Ref37
D. Page 4, line 17; Ref38
E. Page 10, line18; Ref58
F. Page 14, line 25; Ref30

Experimental design

1. The figures have not been organized. There are too many main figures. I think Figure 3 and Figure 4 should be integrated (a phylogenetic tree using WRKY domains from willow, Arabidopsis and poplar). In addition, I suggest use of different colors in different species instead of different groups (like figure 5). Figure 5,6,7,8 and table 2 can be represented as supplementary data.
2. Figure 2 has to include information of physical cluster the authors mentioned at the page 10, line 18-27. It might be better if the authors use different colors for different groups. In addition, the authors has to address the method for defining a cluster.
3. There is no reason to show Ka/Ks ratio as a figure, not a table. I recommend to make this data into table with the gene names of the pairs and ka/ks ratio. This can be represented as supplementary data.
4. The authors should address how define TDs SDs at M&M section.

Validity of the findings

No comments

Additional comments

The authors have used published genome sequence information from willow to mine for WRKY genes. They performed classification and phylogenetic analysis with other species. It has a lot of information, but some major and minor revisions are needed before the manuscript is finally accepted. Also, I strongly recommend the English editing.

·

Basic reporting

WRKY proteins as plant specific TF are a large family of transcription factors, participating in diverse physiological and developmental roles as well as defense response against biotic and abiotic stresses. The authors reports the genome-based survey of WRKY gene family in Salix suchowensis. they identified 85 WRKY genes in the willow genome and showed that the distribution of WRKY genes on chromosomes, phylogenetic analysis, classification of WRKY genes, structural feature, and expression analyses in various tissues. . I believe that this manuscript is very useful and important for further analysis of WRKY’s function and application in woody plants. But there are some problems in this manuscript. It is necessary to correct/edit some point and describe more details to clear up. It also need to improve the quality of tables or figures with suggesting that some could move into supplementary section to get rid of redundancy

Major point
There are informatics figures and tables, which are no doubt to be useful for audience. but some are a kind of redundancy. How about some table move to supple or combine some figures ? For instance, table 2 and table 3 could move to supplementary section Fig 3 and Fig. 4 could be combined (one phylogenetic analysis is enough) or Fig. 4 could move to supplementary section.
Fig 2 quality should be improved. Their resolution of texts like gene names a is too low to see it and it needs to indicate the region of gene clusters which were explained in text (page 10, third paragraph)
In Fig. 7, SsWRKY73 and SsWRKY79 were indicated as duplication gene pair, but there are different clade in phylogenetic tree and showing different gene feature. Please explain it, just mistake ?.
Fig. 8 conserved Motifs of SsWRKY and table 2 could be depicted in supplementary section. The style and format of Fig. 8, especially, are not suitable for main figure as current state.
Page 6 line 7, “http://bio.njfu.edu.cn/willow_chromosome/BuildGff3_Chr.pl” was not accessible through the internet. What happened?
Page 13 line 25-26 and Fig. 6, In table1 and Fig 5, just I founded small no. of WRKY III genes,
I can not fid the value ’21 pairs’ WRKY III genes and need to describe values and all pairs used to estimation of Ka/Ks as supplementary table or sole table.
Page 15 line 11~21, please indicate fig or table for 33 homologus gene pairs, I can not find the value of gene pairs.


Minor point
Page 4 line 12, the sentence needs to be changed more clearly
Page 4,line 22, need to reference about their ecological and economic value and describe importance and the reason of genome wide-screening of WRKY genes
Page 4 line 21, E-value = 1e-3 means their cut-off ? need to explain more clearly
Page 7, line 5, cut-off value e-value e-20 was selected, is there any reason or reference?
Page 7 line 9, resistance for what? Disease or variable biotic ?
Page 8 line 8, Blastp  BLASTP, and when describing cut-off please use the same term through manuscript
Page 8 line 14, RNA-Seq reads were generated in this experiment ? or used them from previous study? Please clear up this
Page 9 line 1 Blastp  BLASTP please use the same term through manuscript
Page 11 line 20-25, after explain about Fig. 4, How about depict these ?

Experimental design

No comments

Validity of the findings

No comments

Additional comments

No comments

---

## Round 0.2 · Minor Revisions

Please revise the following items before you submit final version
 
1.     Line 43-48 (Such as~~): This is not a sentence. This should be connected with the sentence before.
2.     Line 92 (Zhang et al. also identified~~): The reference is “Zhang &Wang”, not “Zhang et al.”
3.     Line 182 (Previous study of Zhang et al. held~~): Same as above. The reference is “Zhang &Wang”, not “Zhang et al.”
4.     Line 200-202 (Gene clusters, defined as~): This sentence should be modified to suit the method section. In addition, “very important for predicting coexpression genes or potential function of clustered genes in angiosperms” is exactly same phrase in the result.
5.     Line 249 (Eulgem et al. previously described~): Same as above. The reference is “Eulgem”, not “Eulgem et al.”
6.     In Fig.2, legend for explaining physical cluster (red line) is missing.
7.     In Fig.3 legend, “T” is missing in front of the legend (he => The)
8.     Table S3 and Table S2 should be changed. Table S3 has been mentioned before Table S2.
9.     Line 570 (inc sensitivity to ~): What does “inc” mean?
10.   Line 573 (so do their orthologs in willow): This phrase is too assertive.

---

## Round 0.3 · accepted · Accept

I think all the reviewers concerns were figured out and the ms is acceptible.